# Oral Health Status and Need for Oral Care in an Aging Population: A Systematic Review

**DOI:** 10.3390/ijerph16224558

**Published:** 2019-11-18

**Authors:** Dorina Lauritano, Giulia Moreo, Fedora Della Vella, Dario Di Stasio, Francesco Carinci, Alberta Lucchese, Massimo Petruzzi

**Affiliations:** 1Department of Medicine and Surgery, Centre of Neuroscience of Milan, University of Milano-Bicocca, 20126 Milan, Italy; moreo.giulia@gmail.com; 2Interdisciplinary Department of Medicine, University of Bari, 70121 Bari, Italy; dellavellaf@gmail.com (F.D.V.); massimo.petruzzi@uniba.it (M.P.); 3Multidisciplinary Department of Medical and Dental Specialties, University of Campania-Luigi Vanvitelli, 80138 Naples, Italy; dario.distasio@unicampania.it (D.D.S.); alberta.lucchese@unicampania.it (A.L.); 4Department of Morphology, Surgery and Experimental medicine, University of Ferrara, 44121 Ferrara, Italy; crc@unife.it

**Keywords:** oral health, aging population, oral disease, association with periodontal disease, oral care need, oral care strategies

## Abstract

Background. The world population is aging. This phenomenon is accompanied by an increase in the number of elderly with dementia, whose oral hygiene care is a challenge. Objective. This paper presents a literature review of oral health status and the need for oral care in people with dementia, as compared to people without dementia and also of the relationship between periodontal disease and cognitive impairment. Methods. A systematic review was conducted in PubMed, CINAHL, and the Cochrane Library. Fifty-six articles met the inclusion criteria and were consequently included for quality assessment and data extraction. Results. No significant differences were found between both groups with regard to the number of present teeth, DMFT Index, edentulousness/use of denture, and orofacial pain. Coronal/root caries and retained roots were more common in people with dementia than in those without dementia. Most of the participants with dementia presented gingival bleeding or inflammation and they suffered from the periodontal disease more than people without dementia. Conclusions. Poor oral health is a common condition among the elderly with dementia. The education process of caregivers might improve the oral health status of people with dementia. Finally, periodontal disease might contribute to the onset or progression of dementia.

## 1. Introduction

The transition from high to low mortality and fertility that accompanied the socioeconomic development of this century has meant a shift in the leading causes of disease and death and an increase of general health problems [1].

As a consequence, a decline of the oral health conditions of the elderly, such as dental caries and periodontal disease, is to be expected [2]. Poor oral health is more common in the elderly suffering from dementia, a disorder that will become prevalent with advancing age of the world population. Several studies analyzed the relationship between poor oral health and cognitive impairment, which suggests that cognitive decline might negatively impact oral health and also that poor oral health might lead to cognitive decline via specific biological mechanisms [3]. Conversely, in a recent systematic review by Wu et al. it has been argued that, according to some studies, it is unclear how or whether oral health conditions and cognitive status are related [4].

In cognitively impaired elderly, the increased incidence of oral disease might be favored by their general conditions: cognitive decline, loss of memory, learning disabilities, attention deficits, and motor skills deterioration, which result in reduced ability to perform routine oral care [5]. A frequent difficulty among these subjects is also the refusal of oral hygiene care, not opening their mouth, using abusive language, or being aggressive.

Even in the early stages of the disease progression, the elderly with dementia have a reduced salivary flow, which can lead not only to a higher prevalence of dental caries [6], but also to difficulties with eating and swallowing, which compromises communication skills [7]. The study by Silva et al. demonstrated that people with dementia living in Australians nursing homes have higher levels of untreated coronal and root caries, due to their cognitive condition and to poor access to professional services [8].

Periodontal disease is higher in the elderly with dementia than in normal cognitive individuals and the periodontal status worsens with the cognitive impairment progression [9]. Evidence of the association between periodontal disease and dementia has been demonstrated: periodontal disease is an inflammatory illness that affects the mouth and it could systemically affect individuals who are vulnerable to dementia and contribute to its pathogenesis [10].

Oral mucosal lesions are frequent in the elderly: the most common denture-related lesions are stomatitis, angular cheilitis, ulcers, hyperplasia, or candidosis. Reduced cognitive independence and decline in self-care, due to dementia, delirium, and social isolation, and it could make oral mucosal condition even worse [11].

Oral health problems in non-verbal individuals could negatively impact life quality, since they cannot communicate their pain and discomfort. On this topic, the study by Merlijn W. de Vries et al. proved the reliability of “the Orofacial Pain Scale for Non-Verbal Individuals” (OPS-NVI) and its psychometric evaluation (Delwel et al. 2018), to establish the presence of pain-related nonverbal communication, such as facial expressions, body movements, and vocal expressions [12].

### 1.1. Rationale of the Systematic Review

The most important risk factor that is associated with the onset of dementia is age and in an aging society, the impact of the phenomenon will be of alarming dimensions. Dementia reduces the quality of life of patients also with regard to oral health. It has been demonstrated that oral health of the elderly with dementia is poorer than oral health in people with normal cognitive function [13,14]. However, it is currently unclear how or whether oral health and cognitive status are related [15,16]. This study is aimed at reviewing literature, in order to evaluate oral health status in elderly with dementia, while comparing the data with those of elderly without dementia and to establish and assess the need of specific oral care strategies, which could improve their quality of life. This research was also conducted to review the available data regarding the influence of periodontal disease on the progression of cognitive impairment.

### 1.2. Objectives

The purpose of this systematic review was to examine studies about oral health in elderly with and without dementia, focusing the research on coronal and roots caries, number of remained teeth and retained roots, Decayed Missing Filled Index, periodontal disease, utilization of dentures, salivary flow, oral hygiene, oral mucosal lesions, orofacial pain, and on the analysis of periodontal disease as a potential risk factor for dementia.

This study reviewed cohort, case-control, cross-sectional studies, and randomized controlled clinical trials in order to examine oral health comparing the elderly with and without dementia. Inclusion criteria of participants were a diagnosis of dementia and the availability of data related to their oral health.

### 1.3. Clinical Question (PICO):

P: A population of participants with diagnosis of dementia aged 60 years or olderI: Analysis of the oral health status and of the association between periodontal disease and dementiaC: Comparison between oral health of elderly with and without dementiaO: Prevalence of oral disease (affecting hard and soft tissues) in elderly with dementia compared to those without dementia, to define their need of oral care. Role of tooth loss due to periodontal disease in the onset/progression of dementia.

## 2. Material and Methods

### 2.1. Protocol and Registration

Methods and inclusion criteria were selected following the PRISMA statement [17], which offers a protocol with respect to the reference items that were included in this systematic review.

### 2.2. Eligibility Criteria

Inclusion and Exclusion Criteria

In this systematic review, all of the articles concerning oral health in the elderly with dementia meeting the following requirements were included:
Participants must have been diagnosed with dementiaQuantitative data about oral health problemsParticipants had to be availableParticipants had to be 60 years or olderCohort, case-control, cross-sectional studies, and clinical trial were considered


The exclusion criteria were as follows:
Case report and reviewsNo quantitative data availableAge of participants below 60 years


#### 2.2.1. Search

An electronic research was conducted to identify relevant studies that have been published within 2019, but no restrictions were imposed with regard to language of the primary studies or methodology. The following electronic databases were used: PubMed, CINAHL, and Cochrane Library. The keywords used were the same for all three databases and they were combined with the Boolean term “AND”: “oral health”, “aging population”, “oral disease”, “association with periodontal disease” and “OR”: “oral care need”, and “oral care strategies”. The research was completed on May 2019.

#### 2.2.2. Study Selection

Two researchers (G.M., D.L.) independently analyzed the title, abstract, and full text of each English article to identify those that were eligible for the systematic review, according to the inclusion and exclusion criteria established above. Disagreements between reviewers were resolved by consensus. Articles that were published in other languages were assessed by a native speaker specialized in medical language. Articles in which the diagnosis of dementia was not defined or quantitative data were not available were excluded.

#### 2.2.3. Data Collection Process

Two reviewers (G.M., D.L.) extracted the data, who also checked their methodological and clinical heterogeneity (D.L.). The information extracted from each article were as follows: study design (cohort, case-control, cross-sectional studies, or randomized clinical trial), participants characteristics, such as age and diagnosis of dementia, and quantitative data on participants (outcome measures), including the number of teeth present, number of retained roots, DMFT Index (Decayed Missing Filled Teeth Index), edentulousness and dentures, coronal and root caries, periodontal health, and its association with the onset or progression of dementia, oral mucosal disease, salivary flow, orofacial pain, oral hygiene, and need for dental treatment. Means and percentages were used for the principal outcome measures.

#### 2.2.4. Quality assessment

Newcastle-Ottawa scale (NOS) [18] was used to assess the quality of the studies (Table 1, Table 2, Table 3 and Table 4). The highest score was 9 and the lowest one was 1 (average score was 5.5 or cohort studies, 6.1 for case-control studies, 4.6 for cross-sectional studies, and 5.5 for RCT). Most articles used the standardized method to examine oral health and the examination was considered to be adequate if a dentist performed it. With regard to “Comparability”, 30 of all the studies controlled for age or gender or both and in the case-control, cross-sectional studies, and RCT only 5 (1.98% of the non-cohort studies) described the non-response rate [14,19,20]. In almost all studies, the duration of follow-up period was longer than three months. The evaluated quality parameters are shown in Additional file 1–4.

## 3. Results

### 3.1. Study Selection and Characteristics

A total of 922 studies that were published between 1990 and 2019 were identified from database searches. Among these articles, after examining titles and abstracts and the full texts of the remaining, only 56 met the inclusion criteria and were consequently included for quality assessment and data extraction. One study was added after scanning the reference list of the included articles [39]. All of the studies were analyzed with regard to quality while using the Newcastle-Ottawa Scale (NOS). Figure 1 shows the flow chart of publication assessment.

The detailed characteristics about the 56 included studies are presented in Table 5, and Table 6, with reference to author and year of publication, study design, case and controls, mean age of participants, dementia measure, and measure of oral health. This review included 19 cohort studies, 9 case-control-studies, 26 cross-sectional studies, and two randomized clinical trials. Almost all of the articles were in English, except for Sumi et al. article, which was written in Japanese [33]. A native Japanese speaker, who extracted the data included in the review, examined this article.

The studies that were selected for the review included in total 8466 participants with dementia and 6797 participants without dementia. In particular, selected studies regarding the association between periodontal disease and dementia included 4698 participants with periodontal disease or history of teeth extraction, 3132 elderly without periodontal disease, 60 subjects with dementia, and 2885 without dementia. In the included studies, the diagnosis of dementia was performed while using DSM-III and IV [70], ICD-9 and 10 (International Classification of Disease [71]), NINCDS-ADRDA (National Institute of Neurological and Communicative Disorders and Stroke and the Alzheimer’s disease and Related Disorders Association [72]), Minimental State examination [73] (MMSE), and other additional measures (e.g., computed tomography, magnetic resonance imaging, CDR [74]).

### 3.2. Results of Individual Studies

The number of present teeth was one of the most used measures for assessing oral health. In particular, from the included studies it came to light that the range within the two groups varied between 2.0 to 20. 2 for people without dementia and between 1.7 to 20.5 for people with dementia. According to Delwel et al. [53], the number of present teeth was lower in the participants with dementia (median = 2.0, IQR = 0.0–18.0) than in people with MCI (median = 18.0, IQR = 5.5–24.0). However, this study underlined that, if only dentate participants were considered, no significant differences were recorded between the two groups (median = 18.0, IQR = 9.0-24-0) (Table 7). The review showed that coronal and root caries were more common in people with dementia that in people without dementia: coronal caries varied between 0.1–2.9 [43,54,55] and 0.0-1-0, respectively [21,54], and root caries varied between 0.6–4.9 [43,54,55] in participants with dementia and 0.3–1.7 [13,54,55] in normal cognitive participants. For retained roots, the range was between 0.0–1.2 [54] in people without dementia and between 0.2–10 [21,54] in people with dementia. Moreover, Delwel et al. [53] demonstrated that dentate participants with dementia had more coronal caries (median = 1.0, IQR = 0.0–2.0), root caries, and retained roots (median = 0.0, IQR = 0.0–1.0) than dentate people with MCI (Table 8. With regards to the DMFT Index (Table 9), the lowest one was 14.9 in the study by Srilapanan et al. In general, the DMFT index did not show a significant difference within the two groups (19.7–26.1 in healthy people and 14.9–28.0 in people with dementia), except for one study, which demonstrated that the DMFT Index was 25.5 in people without dementia and 28.0 in people with dementia.

Most of the participants with dementia presented gingival bleeding or inflammation [23,48] (Table 10). According to De Souza [24], the Gingival Bleeding Index was 46.0% in the elderly with dementia and periodontal infections were most common in the latter (58.6%) than in normal cognitive participants (26.7%). 73.8% of the Delwel et al. [53] study included patients had periodontal pockets of ≥4 mm, 18.8% of them had one or more teeth with mobility grade 2, and 5.8% had one or more teeth with mobility grade 3.

Zenthöfer [36,37,44] demonstrated that the Gingival Bleeding Index of people with dementia was 43.8 to 53.8% and confirmed De Souza’s results, proving that people with dementia suffer from periodontitis more than people without dementia (community periodontal index of treatment needs was 3.1–3.4 in dementia people and 2.7–2.8 in non dementia people [36,37]).

Nine of the included studies found no significant differences between both groups with regards to oral hygiene [21,22,27,45,61,62,64,65,66,67] and five studies demonstrated a higher level of plaque in dementia people [14,36,43,56,65]. The Plaque Index by Silness and Loe was 0.7 in the study by Chalmers et al. [13], 2.5 in the study by Gil-Montoya [14] in the elderly with dementia, and 2.0 in the study by Delwel et al. [53]. Sumi et al. [33] showed a Plaque Index by Quigley and Hein of 1.6. O’Leary Plaque Index was significantly higher in dementia people (90.1%) than in non dementia people [36] (73.3%). Finally, Ribeiro [65] et al. established that the Oral Hygiene Index by Green and Vermillion is higher in participants with dementia (4.5) than in participants without dementia (2.2). A significantly higher Debris Index in people with moderate to severe dementia was found [43].

Furthermore, edentuloussnes was a condition that affected a large percentage of the elderly, in particular 11.6 to 72.7% of the elderly with dementia [66,67] and 14.0 to 70% of the elderly without dementia [38,45]. Within partially or totally edentulous participants, denture utilization varied between these percentages: 17.0–81.8% in normal cognitive people and 5.0 to 100% in people with cognitive impairment [27,31] (Table 11).

Data about orofacial pain were extracted from seven of the included studies [13,24,39,40,45,52]. The percentage of the elderly with dementia suffering from orofacial pain was higher than that of participants without dementia: 7.4 to 21.7%, 6.7 to 18.5%, respectively. The cross-sectional study by Delwel et al. [53] carefully examined the presence of orofacial pain in the elderly with dementia or MCI, while using the OPS-NVI [12] and self reported pain. The OPS-NVI was 4% in rest, 10% during drinking, 19% during chewing, and 22% during oral hygiene care. Pain reported by participants with dementia or MCI was 25.7% overall (Table 12).

The feeling of a dry mouth or xerostomia reached the percentages of 22.0% in people with dementia and 8.4% in people without dementia [33] and it was present in 9.1–45% of the cases and 8.4–20.0% of the controls [40,41,42,43]. Gil-Montoya [58] showed a more drug-induced xerostomia in cases (68.5–72.2%) than in the controls (36.5%).

Oral pathology, such as stomatitis and candidiasis, was most common in the cases than in controls. Chu et al. [38] and other authors [24,39,41] reported a percentage of candidiasis of 3.6–30% for cases and 0.0–5.0% for controls. Furthermore, 18.1–59.1% of cases and 0.0–7.4% of controls showed stomatitis [13,27,40].

In conclusion, with regards to the oral care need, the included studies [13,47,49,50,61,64,67] reported a need of 21% for cleaning teeth and dentures in the elderly with dementia. Chalmers et al. [47] demonstrated that, with an increasing severity of cognitive impairment, there is also an increase of oral care need: the assistance need for cleaning teeth and dentures in severe dementia was 100.0%, as compared to the assistance need in moderate dementia, which was 57.2% (teeth) and 97.3% (dentures). Regarding periodontal disease (Table 13), Dintica et al. [25] and Ide et al. [31] found that the mean change (decrease) in MMSE score due to tooth loss was, respectively, −0.94 to 0.37 (nine months follow-up, adjusted for age, sex, and education) and −3.6 to −0.03 (six months follow-up), establishing a significant association between tooth loss and the progression of cognitive impairment. The crude hazard ratio of dementia according to the number of remaining/lost teeth was 1.6 [10,34]. Tiisanoja et al. [68] demonstrated that subjects with pocket depth ≥4 mm had an increased, but not statistically significant, risk of developing Alzheimer’s Disease (Relative risk: 1.54). Furthermore, Yoo et al. [35] demonstrated that periodontal treatment lead to a significant decrease in the incidence of dementia. The prospective community-based study by Kato et al. [30] showed that the number of natural teeth was significantly associated with an individual’s MMSE score: the percentage of cognitively normal subjects (MMSE scores: 27–30) significantly decreased with a decrease in the number of natural teeth (number of teeth = 5–9, percentage of participants without cognitive impairment = 26.7%; number of teeth = 15–19; and, percentage of normal cognitive participants = 44.8%). Kato et al. also demonstrated that the use of artificial teeth was associated with cognitive function preservation.

## 4. Discussion

The purpose of this systematic review was to examine studies regarding oral health in the elderly with and without dementia and to investigate the relationship between periodontal status and dementia. As reported in a recent review by Delwel et al. [75], the analysis of this study showed no significant differences between the case and controls with regards to the number of present teeth [13,21,29,43,53,56,61,67,69] and to the DMFT Index [27,38,61,65]. However, the DMFT categories separately, “decay”, “missing”, and “filled”, give a better indication of disease and treatment need compared to the index, which reports dental caries history as a whole. Coronal and root caries and retained roots are most common in people with dementia [21,56,65], and this condition might be explained by cognitive and behavioral deterioration, which reduced the ability to perform routine oral care [76]. In the elderly with dementia saliva flow rates decreases [32], eating habits change (more cariogenic food) [51,56,61,77], and motor skills and coordination worsen [65], which leads to a lower chewing and swallowing efficiency [56,77].

Another important point to be considered is the oppositional and aggressive behavior towards oral care and decreased communication skills, which represent barriers to oral hygiene and assistance. These obstacles could be overcome by performing oral care education to caregivers and by increasing dental checks of the elderly with dementia [62,78,79].

Moreover, the recent observational study by Delwel et al. [53] recorded a significant correlation between the cognitive impairment level and the number of present/missing/restored teeth and retained roots, which suggests that dementia could have a negative impact on oral health.

Concerning oral soft tissues, this study confirmed Delwel’s et al. review results [80]: gingival bleeding, periodontal disease, mucosal lesions, and xerostomia were found at higher rates in participants with cognitive impairment. Dry mouth was more common in the elderly, who used medication or had radiotherapy history (head and neck) or autoimmune disease [32].

Approximately the same percentage of the elderly either with or without dementia wore dentures [13,42]. However, people with an advanced dementia degree showed a lower use of dentures, as compared to people with moderate dementia, because of the musculature, salivary flow decrease, and lower tolerance of dentures.

The deterioration of verbal communication skills and the higher prevalence of oral disease might cause a higher suffering to people with dementia, due to orofacial pain [45,50,52,53]. In this systematic review, the new information included concern the potential role of periodontal disease as a risk factor for developing cognitive impairment. In fact, all of the studies included in Table 10 demonstrated an association between teeth loss due to periodontal disease and the onset of dementia [10,25,30,31,34,35,60]. This association is based on biological mechanisms: subjects with periodontal disease or antibodies to periodontal bacterial flora show an increased systemic proinflammatory state, which lead to an increase of the cognitive decline rate [31]. According to Yoo et al., the more teeth are lost, the higher incidence of dementia is. The author also stated that monitoring cognitive status in patients with extensive teeth loss could lead to an early diagnosis of dementia. The significant association between the number of natural teeth and the MMSE score was also confirmed in the study by Kato et al., which suggest that the use of artificial teeth could help to preserve the cognitive function.

### Strengths and Limitations

The most important limitation of this study is the result of the quality assessment of the articles, since more than half of the included studies have a score equal or below 5. The number of high quality studies was low and no homogeneity can be found. For this reason, it was impossible to perform a meta-analysis. In the included studies, several different measures were used to evaluate oral health status and some studies did not distinguish between the elderly with and without dementia.

The main strength of this review is the systematic approach and involvement of a multidisciplinary team (dentists, neuropsychologist, pain specialist). In addition, in almost all of the studies reviewed, oral examinations are structured and standardized, and were carried out by dentists.

## 5. Conclusions

The elderly with dementia show a higher level of plaque, coronal and root caries, retained roots, gingival, and periodontal disease. Further attention is needed regarding orofacial pain, which is very common in dementia people. Poor oral health within this group could be increased by the reduction of submandibular salivary flow, deterioration of cognitive functions, motor and communication skills, and aggressive behavior. Caregivers should be educated and dentist’s checks in nursing homes should be enhanced in order to improve the oral health status of the elderly with dementia. Our contribution highlights the relationship between periodontal disease and dementia. Teeth loss due to periodontal disease increases the risk of cognitive function deterioration. However, the specific mechanisms of this association need further investigation.

## Figures and Tables

**Figure 1 ijerph-16-04558-f001:**
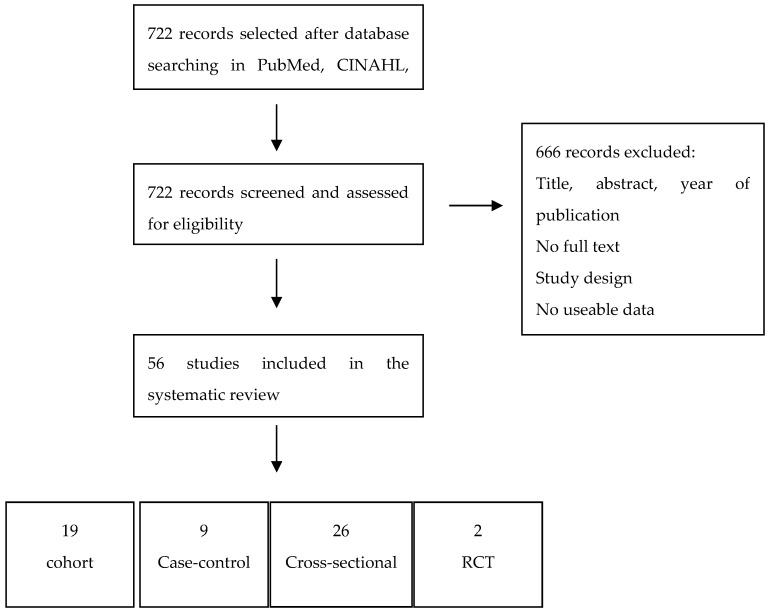
PRISMA flow diagram.

**Table 1 ijerph-16-04558-t001:** Quality assessment for cohort studies (Additional file 1).

Cohort Studies	Representativeness of Exposed Cohort	Selection of Non Exposed Cohort	Ascertainment of Exposure	Demonstration Outcome Not Present at Start of Study	Comparability	Outcome	Total
Chalmers et al. 2002 [21]	+	−	−	−	++	++−	5
Chalmers et al. 2003 [13]	+	−	−	+	++	+++	7
Chalmers et al. 2004 [22]	+	−	−	−	−−	++−	3
Chen et al. 2010 [23]	+	+	+	−	−−	+++	5
De Souza Rolim et al. 2014 [24]	+	−	+	−	−−	+−−	3
Dintica et al. 2018 [25]	+	−	+	+	++	+++	8
Ellefsen et al. 2009 [26]	+	+	+	−	++	++−	7
Hatipoglu et al. 2011 [27]	−	−	−	+	−−	++	3
Hoben et al. 2016 [28]	+	−	+	−	+−	++	5
Jones et al. 1993 [29]	+	−	−	+	++	++−	6
Kato et al. 2019 [30]	+	−	+	+	+−	−++	6
Ide et al. 2016 [31]	−	−	+	+	+−	+++	6
Lee et al. 2017 [10]	+	+	+	−	++	+++	8
Ship and Puckett 1994 [32]	+	+	+	−	+−	++−	6
Sumi et al. 2012 [33]	+	−	+	−	−−	++	4
Takeuchi et al. 2017 [34]	+	−	+	−	++	+++	7
Yoo et al. 2019 [35]	+	−	+	+	++	++−	7
Zenthofer et al. 2014 [36]	−	−	+	+	−−	+++	5
Zenthofer et al. 2016 [37]	+	+	+	+	++	+++	5

+ = star assigned; − = star not assigned.

**Table 2 ijerph-16-04558-t002:** Quality assessment for case-control studies (Additional file 2).

Case-Control Studies	Definition of Cases	Representativeness of Cases	Selection of Controls	Definition of Controls	Comparability	Exposure	Total
Chu et al. 2015 [38]	−	+	−	+	++	+−−	5
De Souza Rolim et al. 2014 [39]	+	+	+	+	++	++−	8
Gil–Montoya et al. 2017 [14]	+	+	−	+	++	−++	7
Hoeksema et al. 2016 [19]	+	+	−	+	−−	−++	5
Kossioni et al. 2012 [40]	+	+	−	−	++	++−	6
Leal et al. 2010 [41]	−	+	−	+	−−	−++	4
Nordenram et al. 1996 [42]	+	+	+	+	++	++−	8
Warren et al. 1997 [43]	+	+	−	−	++	+++	7
Zenthofer et al. 2017 [44]	+	+	+	−	++	−−+	6

+ = star assigned; − = star not assigned.

**Table 3 ijerph-16-04558-t003:** Quality assessment for cross-sectional studies (Additional file 3).

Cross-Sectional Studies	Representativeness of the Sample	Sample Size	Non-Respondents	Ascertainment of the Exposure	Comparability	Outcome	Total
Adam et al. 2006 [45]	−	+	+	−	−−-	+−	3
Bomfin et al. 2013 [46]	−	−	+	−	−−	−	1
Chalmers et al. 2002 [47]	−	+	+	−	−−	++	4
Chapman et al. 1991 [48]	−	+	−	−	−−	+−−	2
Chen et al. 2013 [49]	+	+	+	+	−−	++−	6
Chen et al. 2013-2015 [50,51]	+	+	+	+	−−	−+	5
Cohen-Mansfield 2002 [52]	−	−	−	−	−−	+−−	1
Delwel et al. 2019 [53]	+	+	+	+	++	+++	9
Ellefsen et al. 2008-2012 [54,55]	+	+	+	−	++	++−	7
Elsig et. Al. 2015 [56]	+	+	+	−	−−	++−	5
Eshkoor et al. 2014 [57]	−	+	−	−	−−	−−−	1
Gil-Montoya et al. 2016 [58]	+	+	+	+	−−	++	6
Kossioni et al. 2013 [59]	+	−	+	−−	++	−+−	5
Furuta et al. 2013 [60]	−	+	−	−	++	++−	5
Hopcraft et al. 2012 [61]	−	+	+	−	−−	++−	4
Lee et al. 2013 [62]	+	−	−	+	+−	++	5
Luo et al. 2015 [20]	+	+	+	+	−+	−++	7
Minakuchi et al. 2006 [63]	+	+	−	+	+−	++−	6
Philip et al. 2012 [64]	−	−	−	+	−−	++−	3
Ribeiro et al. 2012 [65]	+	+	−		−−	++−	5
Srilapanan et al. 2013 [66]	−	+	−	−	−−	+−−	2
Syrjala et al. 2012 [67]	+	+	+	+	++	++−	8
Tiisanoja et al. 2018 [68]	−	+	−	+	+	++	5

+ = star assigned; − = star not assigned.

**Table 4 ijerph-16-04558-t004:** Quality assessment for RCT (Additional file 4).

Randomized Clinical Trial	Definition	Representativeness of Cases	Selection of Controls	Definition of Controls	Comparability	Exposure	Total
Fjeld et al. 2014 [69]	+	−	+	+		+++	6
Zenthofer et al. 2016 [44]	+	+	−	−	++	+−−	5

+ = star assigned; − = star not assigned.

**Table 5 ijerph-16-04558-t005:** List of studies about coronal/root caries, number of remained teeth/retained roots, DMFT, periodontal disease, utilization of dentures, salivary flow, oral hygiene, and oral mucosal lesions in elderly with and without dementia.

Study	Design	Dementia Group	Mean Age	Control Group	Mean Age	Dementia and Oral Health Measure
Adam et al. 2006 [45]	Cross-sectional	81	80.8	54	85.5	Abbreviated Mental Test. Orofacial pain, dentures, edentulousness, DMFT, Debris Index
Bonfim et al. 2013 [46]	Cross-sectional	?	?	?	?	MMSE, chart, ADL. Number of present teeth, dentures
Chalmers et al. 2002 [21]	Cohort	116	91: <79 years 25: 80+ years	116	91: <79 years 25: 80+ years	MMSE. Number of present teeth, DMFT, root caries, Plaque Index
Chalmers et al. 2002 [47]	Cross-sectional	224	83.2	0	-	MMSE. Number of present teeth, DMFT, dentures, dental habits, coronal and root caries, retained roots
Chalmers et al. 2003 [13]	Cohort	103	82: <79 years 21:80+ years	113	88:<79 years 25:80+ years	MMSE. Number of present teeth, DMFT, coronal and root caries, Plaque Index
Chalmers et al. 2004 [22]	Cohort	224	83.2	0	-	MMSE. Orofacial pain, number of present teeth, DMFT, dental habits, number of coronal and root caries, retained roots
Chapman et al. 1991 [48]	Cross-sectional	85	74.9	0	-	Dementia measure not described. Number of present teeth, dentures, DMFT, Presence of deposits
Chen et al. 2010 [23]	Cohort	119	81.5	372	73.8	Chart. Present of calculus, plaque and gingival bleeding
Chen et al. 2013 [49]	Cross-sectional	Community: 51 Assisted living: 18 NHR: 501	79.3	0	-	Chart, ICD-9. Number of present teeth, decay or retained roots, presence of calculus, plaque and gingival bleeding, need for oral care
Chen et al. 2013 [50]	Cross-sectional	501	82.6	199	76.1	Chart. Presence of calculus, plaque and gingival bleeding
Chen et al. 2015 [51]	Cross-sectional	46	79.3	138	71.6	Chart. Need for oral care
Chu et al. 2015 [38]	Case-control	59	79.8	59	79.8	Chart. DMFT, Community Periodontal Index, Salivary flow
Cohen-Mansfield 2002 [52]	Cross-sectional	21	88.0	0	-	MMSE, MDS-COGS. Number of broken or fractured teeth, caries, retained roots, dentures, presence of gingivitis and periodontal disease
De Souza Rolim et al. 2014 [24]	Case-control	29	75.2	30	61.2	NINCDS-ADRDA for AD, MMSE. Orofacial pain, DMFT, Gingival Bleeding Index, probing pocket depth, CAL, Plaque Index
De Souza Rolim et al. 2014 [39]	Case-control	29	75.2	0	-	NINCDS-ADRDA for AD, MMS Orofacial pain, DMFT, Gingival Bleeding Index, probing pocket depth, CAL, Plaque Index
Delwel et al. 2019 [53]	Cross-sectional	Dementia:303 MCI: 45		-	-	MMS. Presence of orofacial pain using the OPS-NVI and self-reported pain, number of missing or restored teeth, coronal and root caries, retained roots, DPSI, Plaque Index of Silness and Loë, oral hygiene, oral mucosal lesions
Ellefsen et al. 2008 [54]	Cross-sectional	AD:61 OD:26	82.8 81.5	19	79.8	ICD-10. Coronal caries, root caries
Ellefsen et al. 2009 [26]	Cohort	AD:49 OD:15	83.6 81.3	13	79.9	ICD-10. Number of present teeth, DMFT, CCI, NCI, ADJCI
Ellefsen et al. 2012 [55]	Cross-sectional	61	82.8	0	-	Number of present teeth, DMFT, coronal caries, root caries
Elsig et al. 2013 [56]	Cross-sectional	29	82.5	22	81.9	NPT, MMSE, CERAD, CDR. Number of present teeth, presence of visible dental plaque
Eshkoor et al. 2014 [57]	Cross-sectional	1210	71.0	0	-	MMSE. Presence of teeth or dentures
Fjeld et al. 2014 [69]	RCT	159	85.5	43	88.5	Evaluated by physician. Number of present teeth, Simplified Oral Hygiene Index, mouth dryness
Furuta et al. 2013 [60]	Cross-sectional	204	-	82	-	CDR. Number of present teeth, dentures
Gil-Montoya et al. 2016 [58]	Cross-sectional	MiD:73 MoD:66 SeD:36	76.4 77.6 80.4	156	77.4	NINCDS-ADRDA. Drug-induced xerostomia
Gil-Montoya et al. 2017 [14]	Case-control	133	80.0	324	79.8	DSM-IVR, NINCDS-ADRDA. Plaque Index, Bleeding Index
Hatipoglu et al. 2011 [27]	Prospective-cohort	31	67.6	47	65.3	MMSE. Dentures, DMFT, oral hygiene status, mucosal pathology
Hoben et al. 2016 [28]	cohort	1606	85.0	1105	83.4	Chart. Presence of debris, of inflamed, swollen or bleeding gums, oral care by staff
Hoeksema et al. 2016 [19]	Case-control	479	84.0	246	81.1	MMSE. Visual plaque according to the score of Mombelli
Hopcraft et al. 2012 [61]	Cross-sectional	510	?	0	-	Chart. Number of present teeth, DMFT, coronal caries, retained roots
Ide et al. 2016 [31]	Cohort	59	77.6	0	-	NINCDS-ADRDA. Presence of visible plaque, bleeding on probing, pocket depth, presence of moderate and severe periodontal disease (CDC/AAP criteria)
Jones et al. 1993 [29]	Cohort	23	67.4	46	66.1	Longitudinal study of dementia. Present Teeth, CCI, RCI
Kossioni et al. 2012 [40]	Case-control	27	76.5	84	-	DSM-IV. Number of present teeth, DMFT, presence of plaque or calculus, mucosal pathology
Kossioni et al. 2013 [59]	Cross-sectional	23	76.3	0	-	Mentally ill, including dementia. Mucosal complaints
Leal et al. 2010 [41]	Case-control	20	69.6	20	68.3	NPT, CDR, MMSE. Mucosal pathology, salivary flow
Lee et al. 2013 [62]	Cross-sectional	19	83.9	169	77.4	MCI, MiD, DSM-IV. Missing teeth, coronal and root caries, periodontal pocket depth, Plaque Index
Luo et al. 2015 [20]	Cross-sectional	120	80.9	2389	70.0	DSM-IV. Missing teeth
Minakuchi et al. 2006 [63]	Cross-sectional	155		50		COD by MHLW JP. Number of present teeth, dentures
Nordenram et al. 1996 [42]	Case-control	40	87.0	40	87.0	DSM-III-R, MMSE. Number of present teeth, dentures
Philip et al. 2012 [64]	Cross-sectional	84	85.7	102	84.3	Chart, ADLOH. DMFT, retained roots, degree of gingival inflammation, Plaque Index
Ribeiro et al. 2012 [65]	Cross-sectional	30	79.1	30	67.8	ICD-10, DSM-IV, MMSE, CDR. Number of present teeth, dentures, DMFT, Oral Health Index
Ship et al. 1994 [32]	Cohort	21	64.0	21	65.0	NINCDS-ADRDA, CT, MRI, PET, PT. Number of present teeth, DMFT, Change in gingival status, periodontal pocket depth, salivary flow
Srilapanan et al. 2013 [66]	Cross-sectional	69	75.5	0	-	Chart, MMSE. Dental Habits, number of present teeth, dentures, DMFT, caries, Community Periodontal Index, need for oral care
Sumi et al. 2012 [33]	Cohort	10	77.7	0	-	NINCD-ADRDA, MMSE. Number of present teeth, DMFT, Gingival Index, Plaque Index
Syrjala et al. 2012 [67]	Cross-sectional	49	84.8	278	81.4	DSM-III-R, DSM-IV, McKeith. Number of present teeth, dentures, number of teeth with periodontal pockets > 4 mm, presence of poor oral hygiene
Warren et al. 1997 [43]	Case-control	45	81.6	133	80.3	MMSE, chart, NT, scans. Number of present teeth, dental habits, coronal and root caries, dentures, modification of Gingival Index, modification of Debris Index, xerostomia, mucosal pathology
Zenthöfer et al. 2014 [36]	Cohort	57	83.1	36	82.6	MMSE, chart. Decayed and missing teeth, periodontitis, Gingival Bleeding Index, CPITN, Dental Hygiene Index, Plaque Control Record
Zenthöfer et al. 2016 [37]	Cohort	33	81.7	60	83.4	MMSE, chart. Missing teeth, Gingival Bleeding Index, CPITN, Plaque Control Record, mucosa pathology
Zenthöfer et al. 2016 [44]	RCT	136	84.6	83	80.7	MMSE. Gingival Bleeding Index, CPITN
Zenthöfer et al. 2017 [44]	Case-control	136	84.6	83	80.7	MMSE. Gingival Bleeding Index, CPITN

ADJCI: adjusted caries increment; CAL: clinical attachment level; CCI: crude caries increment; CDC/AAP: Center for Disease Control/American Academy of Periodontology; COD: classification of dementia; CPITN: community periodontal index of treatment needs CT: computer tomography; DMFT: decayed missing filled teeth; DPSI: Dutch Periodontal Screening Index; ICD: International Classification of Disease; MCI: Mild Cognitive Impairment; Mckeith: Consensus criteria presented by McKeith; MDS-COGS: Minimum Data Set Cognition Scale; MHLW: Ministry of Health, Labour and Welfare; MMSE: Mini Mental State Examination; MRI: magnetic resonance imaging; NCI: net caries increment NINCDS-ADRDA: National Institute of Neurological Disorders and Stroke Alzheimer’s Disease and Related Disorders Association; NHR: nursing home residents; NPT: Neuropsychological testing; NT: Neurological testing; OPS-NVI: ORrofacial Pain Scale for Nonverbal Individuals; PET. Positron emission tomography.

**Table 6 ijerph-16-04558-t006:** List of studies about the relationship between dementia and periodontal disease.

Study	Design	Dementia or Periodontal Disease Group	Mean Age	Control Group	Mean Age	Dementia and Oral Periodontal Disease Measure:
Dintica et al. 2018 [25]	Cohort	Free D: 2715	-	-		Dementia incidence: DSM-IV, MMSE.
Ide et al. 2016 [31]	Cohort	Mid to Mod: 60 30 men 30 women	77.7	-	-	Presence or absence of periodontal disease following CDC/AAP
Kato et al. 2019 [30]	Cohort	210 Men: 103 Women: 107	78.1 +− 4.9	-	-	Cochran-Armitage trend test Mann-Whitney U test Kruskal-Wallis test Spearman’s rank correlation test Student’s *t*-test
Lee et al. 2017 [10]	Cohort	PD: 3028 (54% male)	72.4	No PD: 3028 (54% male)	72.4	Dementia incidence: ICD-9-CM codes 523.3–5
Takeuchi et al. 2017 [34]	Cohort	PD: 1566 (691 men and 875 women)	-	-	-	Dementia incidence: DSM-III, MMSE, HDS-R.
Tiisanoja et al. 2018 [68]	Cross-sectional	IG: 170	80.9 AD: 82.2 OD: 82.5	-	-	DSM-IV, McKeith et al. MMSE. Structured oral examination examining number of teeth with periodontal pocket depth of 4 mm or more
Yoo et al. 2019 [35]	Cohort	TE: 104,903	-	NE: 104,903	-	Dementia incidence: Disease codes and definition codes associated with dementia

AD: Alzheimer’s disease; CDC/AAP: Centre for Disease Control/American Academy of Periodontology; FreeD: Free-dementia; DI: Dementia incidence; HDS-R: Revised Hasegawa’s Dementia Scale; IG: Intervention Group; Mid to Mod: mild to moderate dementia; OD: other dementia; PD: Periodontal Disease; NHANES: National Health and Nutrition Examination Survey; NE: non-extraction cohort; TE: tooth extraction.

**Table 7 ijerph-16-04558-t007:** Results about the number of present teeth.

Study	Number of Present Teeth No Dementia	Number of Present Teeth Dementia	*p* Value: Dementia vs. No Dementia
Bomfin et al. 2013 [46]	2.0	3.0	-
Chalmers et al. 2002, 2003 [13,21,47]	17.2	18.0	>0.05
Chapman et al. 1991 [48]	-	12.8	-
Chen et al. 2010 [23]	-	18.2 community living 19.3 assisted living	-
Delwel et al. 2019 [53]	-	Dementia:18.0 MCI: 22.0	0.058
Ellefsen et al. 2009 [26]	20.2	17.3 AD 16.1 OD	≤ 0.001
Ellefsen et al. 2012 [55]	-	16.5	-
Elsig et al. 2013 [56]	6.5	4.9	0.533
Fjeld et al. 2013 [69]	20.1	20.0	-
Hopcraft et al. 2012 [61]	14.6	0.7	>0.05
Jones et al. 1993 [29]	18.2	17.9	0.90
Kossioni et al. 2012 [40]	-	4.4	-
Ribeiro et al. 2012 [65]	Median 13.5 (0.0-28-0)	Median 1.0 (o.o-22-0)	0.0004
Srisilapanan et al. 2013 [66]	-	19.5	-
Sumi et al. 2012 [33]	-	12.7	-
Syrjala et al. 2012 [67]	15.0	10.9 AD 7.8 VaD 1.7 OD	-
Warren et al. 1997 [43]	13.0	10.0 AD 13.0 OD	*p* > 0.05

AD: Alzheimer dementia; OD: other dementia; Vad: vascular dementia.

**Table 8 ijerph-16-04558-t008:** Results about coronal/root caries and retained roots.

Study	Coronal Caries No Dementia	Coronal Caries Dementia	Root caries no Dementia	Root Caries Dementia	Retained Roots no Dementia	Retained Root Dementia
Chalmers et al. 2002–2003 [13,47]	0.0 *	* 0 ** 5	0.3	0.8	Decayed: 0.0 * Sound: 0.1	Decayed: 0.3 * Sound: 0.1
De Souza Rolim et al. 2014 [24,39]	3.4%	6.8%	-	-	10.2%	6.8%
Delwel et al. 2019 [53]	-	27.0%	-	19.0%	-	18.1%
Ellefsen et al. 2008–2012 [54,55]	1.0 *	2.9 *	1.7 *	4.9 * AD 2.3 * OD	0.0	AD 10.0 * OD 0.5 *
Jones et al. 1993 [29]	0.8	1.4	0.4	1.8	-	-
Lee et al. 2013 [62]	0.8	1.0	0.5 *	1.8 *	-	-
Philip et al. 2012 [64]	-	-	-	-	1.2	1.8
Warren et al. 1997 [43]	0.4	AD 0.1 OD 0.4	0.8	AD O.6 OD 0.6	-	-

AD: Alzheimer’s disease; OD: other dementia; *: *p* ≤ 0.05; **: *p* ≤ 0.01.

**Table 9 ijerph-16-04558-t009:** Results about the DMFT Index.

Study	DMFT No Dementia	DMFT Dementia
Adam et al. 2006 [45]	Decayed 1.1 Missing 28.2 Filled 0.7	Decayed 0.80 Missing 27.3 Filled 0.90
Chalmers et al. 2002 [21,47]	Decayed 0.0–0.4 Missing – Filled 24.7–25.7	Decayed 0.5–1.6 * Missing – Filled 22.1-23.9
Chalmers et al. 2003 [13]	Decayed 0.0–0.1 Missing - Filled -	Decayed 0.3–1.3 * Missing - Filled -
Chapman et al. 1991 [48]	Decayed - Missing - Filled -	Decayed 1.4 Missing 17.8 Filled 6.4 DMFT 25.6
Chen et al. 2013 [49,50]	Decayed - Missing - Filled -	Decayed 5.5 (C), 5.3 (A), 6.0 (NHR) Missing - Filled 10.4 (C), 10.9 (A), 8.7 (NHR)
Chu et al. 2015 [38]	Decayed 0.8 Missing 18.3 Filled 2.4 DMFT 21.5	Decayed 1.2 Missing 18.9 Filled 2.5 DMFT 22.3
De Souza Rolim et al. 2014 [24,39]	Decayed - Missing - Filled -	Decayed - Missing - Filled – DMFT 27.2 Range 11–32
Hatipoglu et al. 2011 [27]	Decayed Missing Filled DMFT 19.7	Decayed Missing Filled DMFT 24.2
Hopcraft et al. 2012 [61]	Decayed 2.9 Missing 17.4 Filled 4.8 DMFT 25.0	Decayed 2.4 Missing 17.9 Filled 4.8 DMFT 25.0
Kossioni et al. 2012 [40]	Decayed - Missing - Filled -	Decayed 1.8 Missing - Filled 0.9
Lee et al. 2013 [62]	Decayed CC+RC Missing 12.7 Filled -	Decayed CC+RC Missing 10.2 Filled -
Luo et al. 2015 [20]	Decayed - Missing 9.3 Filled -	Decayed - Missing 18.7 ** Filled -
Philip et al. 2012 [64]	Decayed 2.9 Missing 18.0 Filled 5.0 DMFT 26.1	Decayed 3.0 Missing 17.4 Filled 5.3 DMFT 25.9
Ribeiro et al. 2012 [65]	Decayed Missing Filled DMFT 25.5 *	Decayed Missing Filled DMFT 28.0 *
Srisilapanan et al. 2013 [66]	Decayed - Missing - Filled -	Decayed 1.5 Missing 12.6 Filled 0.8 DMFT 14.9
Zenthöfer et al. 2014 [36]	Decayed 0.7 Missing 19.9 Filled -	Decayed 0.6 Missing 20.8 Filled -
Zenthöfer et al. 2016 [37]	Decayed - Missing 20.5 Filled -	Decayed - Missing 20.5 Filled -

CC: coronal caries; DMFT: decayed missing filled teeth; NHR: nursing homes residents; RC: root caries; *: *p* ≤ 0.05; **: *p* ≤ 0.01.

**Table 10 ijerph-16-04558-t010:** Results about gingival and periodontal disease.

Study	Outcome Measure	Gingival Health No Dementia	Gingival Health Dementia
Chen et al. 2010 [23]	Calculus/plaque/gingival bleeding	No: 1.2% M to M: 85.5% High: 13.3%	No: 0.9% M to M: 67.9% High: 31.3%
Chen et al. 2013 [49]	Calculus/plaque/gingival bleeding	-	C: 0% No, 65.8% Small, 34.2 High A: 8.3% No, 66.7% Small, 25.0% High NHR: 59.2% No, 59.2% Small, 40.5% High
Chen et al. 2013 [50]	Calculus/plaque/gingival bleeding	No:0.0% M to M: 73.8% ** High: 26.2%	No: 0.3% M to M: 59.2% High:40.4%
Chu et al. 2015 [38]	Community Periodontal Index, pockets ≥ 3 mm	74.0%	78.0%
Cohen-Mansfield et al. 2002 [52]	Periodontal disease Gingivitis	-	44.4% 38.9%
De Souza Rolim et al. 2014 [24]	Probing pocket depth (mm) Gingivitis MoD-SeV periodontitis Periodontal infection Gingival Bleeding Index	- 10.0%-10.0% 6.7% 26.7% -	1.6 mm 31.0% 6.9%–20.7% 58.6 ** 46.0%
De Souza Rolim et al. 2014 [39]	Probing pocket depth (mm) Gingivitis MoD-SeV periodontitis Periodontal infection Gingival Bleeding Index	- 10.0%–10.0% 6.7% 26.7% -	1.6 mm 31.0% 6.9%–20.7% 58.6 ** 46.0%
Delwel et al. 2019 [53]	Silness and Loë Plaque Index DPSI Probing pocket depth (mm) Mobility grade		2.0 3- (IQR = 2.4) ≥4 mm in 73.8% participants Grade 2 in18.8% Grade 3 in 5.8%
Gil-Montoya et al. 2016 [58]	Bleeding Index	50.6	67.5 ***
Hoben et al. 2016 [28]	Inflamed, swollen or bleeding gums	1.2%	0.8%
Hopcraft et al. 2012 [61]	Periodontal pocket depth 4 mm, periodontal pocket depth > 6 mm	35.0%	36.0%
Ide et al. 2016 [31]	Probing depth > 3 mm Bleeding on probing Periodontitis according to CDC/AAP criteria: MoD-SeV		6.7% 13.6% 37.3 25.4%–1.9%
Lee et al. 2013 [62]	Periodontal pocket depth	1.5 mm	1.4 mm
Philip et al. 2012 [64]	Gingival inflammation: Minimal: Light: Moderate:	22.5% 36.3% 32.5% 6.8%	13.0% 21.0% 56.5% 13.0%
Sumi et al. 2012 [33]	Gingival Index Loe-Silness	-	1.2
Srisilapanan et al. 2013 [66]	Community Periodontal Index, highest score: Normal Bleeding Calculus Pocket depth 4–5 mm Pocket depth ≥ 6 mm	- - - - -	9.4% 1.9% 34.0% 30.2% 24.5%
Syrjala et al. 2012 [67]	Number of teeth with periodontal pockets		2.8 AD 2.8 VaD 1.7 OD
Warren et al. 1997 [43]	Modified version of Gingival Index by Silness and Loe	0.7	1.1 AD 1.2 0.9 OD
Zenthöfer et al. 2014 [36]	Periodontitis Gingival Bleeding Index Community periodontal index of treatment needs	73.9% 40.9% 2.8	100% *** 43.8% 3.4 ***
Zenthöfer et al. 2016 [37]	Gingival Bleeding Index Community periodontal index of treatment needs	38.1% 3.1	52.1% * 3.3
Zenthöfer et al. 2016 [44]	Community periodontal index of treatment needs	2.7	3.1 ***
Zenthöfer et al. 2017 [44]	Gingival Bleeding Index Community periodontal index of treatment needs	48.8% 2.7	53.8% 3.1 ***

AD: Alzheimer’s disease; Vad: vascular dementia; OD: other dementia; *: *p* ≤ 0.05; **: *p* ≤ 0.01; ***: *p* ≤ 0.00.

**Table 11 ijerph-16-04558-t011:** Results about dentures and edentulousness.

Study	Dentures no Dementia	Dentures Dementia	Edentulousness No Dementia	Edentulousness Dementia
Adam et al. 2006 [45]	/	/	70.0%	63.0%
Bomfim et al. 2013 [46]	20.0%	20.0%	46.7%	40.0%
Chalmers et al. 2003 [13]	27.6-30.1%	20.7-23.3%	/	/
Chapman et al. 1991 [48]	-	59.0%	-	64.7%
Chen et al. 2013 [49]	-	48.0% C 38.9% A 47.1% NHR	/	/
Chu et al. 2015 [38]	14.0%	17.0%	/	/
De Souza Rolim et al. 2014 [24]	43.3%	25.8%	43.3%	32.3%
Elsig et al. 2013 [56]	/	/	54.6%	62.1%
Eshkoor et al. 2014 [57]	81.8%	86.2%	/	/
Hatipoglu et al. 2011 [27]	57.0% Max 55.0% Mand	97.0% Max 100.0% Mand	/	/
Kossioni et al. 2012 [40]	-	62.9%		
Nordenram et al. 1996 [42]	17.0%	7.0% Mod 5.0% Sed	43.0%	36.0% MoD 45.0% SeD
Ship et al. 1994 [32]	43.0%	40.0-67.0%	/	/
Srisilapapanan et al. 2013 [66]	-	40.6%	-	11.6%
Syrjala et al. 2012 [67]	73.7%	75.5% AD 68.6% VaD 72.2% OD	44.6%	63.3% AD 68.8% VaD 72.7% OD
Warren et al. 1997 [43]	/	/	31.6%	40.0% AD 32.0% OD

Mand. Mandibular; Max: Maxillary; MoD: moderate dementia; NHR: nursing home residents; SeD: severe dementia; Vad: vascular dementia; OD: other dementia; *: *p* < 0.05.

**Table 12 ijerph-16-04558-t012:** Results about Orofacial pain.

Study	Orofacial Pain No Dementia	Orofacial Pain Dementia
Adam et al. 2006 [45]	18.5%	7.4%
Chalmers et al. 2003 [13]	11.2–11.5%	18.4–19.0%
Cohen-Mansfield et al. 2002 [52]	-	60.0%
De Souza Rolim et al. 2014 [24]	6.7%	20.7%
Delwel et al. 2019 [53]	-	Dementia: 27.4% MCI: 20.5%
Kossioni et al. 2012 [40]	-	21.7%

MCI: Mild Cognitive Impairment.

**Table 13 ijerph-16-04558-t013:** Association between periodontal disease and dementia.

Study	Follow-Up Period	Association between Tooth Loss/Periodontal Disease and Dementia
Dintica et al. 2018 [25]	9 years	Annual mean change in MMSE by tooth loss: −0.94 to −0.37 *
Ide et al. 2016 [31]	6 months	Mean change in MMSE by the presence of periodontal disease: −3.6 to −0.03 −3.6 to 0.04 ***
Kato et al. 2019 [30]	4 years	Cochran-Armitage trend test Mann-Whitney U test Kruskal-Wallis test Spearman’s rank correlation test Student’s *t*-test
Lee et al. 2017 [10]	Index date: date of the first periodontal disease diagnosis. Patients were followed until dementia diagnosis, death, withdrawal from the NHI or December 31, 2012	Hazard ratio of having dementia in persons with periodontal disease → HR: 1.02–1.32 1.01–1.32 **
Takeuchi et al. 2017 [34]	5 years	Hazard ratio of having dementia in persons with 10-19 remaining teeth → HR: 1.45–3.03 1.06–2.46 ***
Tiisanoja et al. 2018 [68]		Risk of having AD in persons with pocket depth ≥ 4 mm → RR: 1.54
Yoo et al. 2019 [35]	7 year	Odds Ratio of having dementia in persons with 7–12 teeth lost → OR: 1.272

HR: hazard ratio; *: adjusted for age, sex and education; **: adjusted for sociodemographic characteristics and comorbidities; ***: adjusted for baseline age, gender and cognitive score; NE: non-extraction cohort; NHI: National Health Insurance; OR: odds ratio; RR: relative risk; TE: tooth extraction.

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
