# Peer review of "Oral Health Status and Need for Oral Care in an Aging Population: A Systematic Review"

_ijerph, 2019, doi:10.3390/ijerph16224558_

Round 1
Reviewer 1 Report
The authors attended, in my opinion, all the suggestions.
Author Response
Milan 05/11/2019
Dear editor,
Many thanks for the insightful comments and suggestions of the referees. We have
made corresponding revisions according to their advice. Words in red are the changes
we have made in the text. The language of the manuscript has also been extensively
revised by a professional MDPI English language science editing service and all authors of
this article have seen and approved the changes.
The revisions are as follows:
REVIEWER 1
The manuscript was modified correctly. There are minor comments as follows;
Please unify the expression. For example, in table 7, "Number of present teeth without dementia" and "Number of present teeth in dementia". On the contrary, in Table 9, "Root caries no dementia" and "Root caries dementia". In Table 2, one line is required between the indices and actual data.
Thank you for your indications.
We have unified the expressions in Table 7-8.
We have added one line between the indices and actual data in Table 2
Thank you for receiving our manuscript and considering it for publication.
We appreciate your time and look forward to your response.
Yours sincerely,
Dorina Lauritano

Reviewer 2 Report
The manuscript was modified correctly. There are minor comments as follows;
Please unify the expression. For example, in able 7, "Number of present teeth without dementia" and "Number of present teeth in dementia". On the contrary, in Table 9, "Root caries no dementia" and "Root caries dementia". In Table 2, one line is required between the indices and actual data.Author Response
Milan 05/11/2019
Dear editor,
Many thanks for the insightful comments and suggestions of the referees. We have
made corresponding revisions according to their advice. Words in red are the changes
we have made in the text. The language of the manuscript has also been extensively
revised by a professional MDPI English language science editing service and all authors of
this article have seen and approved the changes.
The revisions are as follows:
REVIEWER 2
Comments and Suggestions for Authors
The authors attended, in my opinion, all the suggestions.
Thank you for your kind suggestions and indications.
Thank you for receiving our manuscript and considering it for publication.
We appreciate your time and look forward to your response.
Yours sincerely,
Dorina Lauritano

This manuscript is a resubmission of an earlier submission. The following is a list of the peer review reports and author responses from that submission.
Round 1
Reviewer 1 Report
The manuscript is correctly written. The methodology is adequate and
the methodological rigor to perform the systematic search is correct.
PICO is well designed as well as the PRISMA flow-diagram.
Authors are strongly recommended to adjust the title of the manuscript according: the research objectives, methodology used, reference / contextual framework in the introduction, and consequently the conclusions address the issue proposed in the title.
The quality of life associated with oral health is own concept that has its measuring instruments such as OHIP-49/14. This instrument establishes the impact of oral health on the quality of life and it is measured in 7 dimensions. However, these instruments were not included in the systematic review of literature.
Reviewer 2 Report
RE: “Oral health-related quality of life and associated factors in elderly people: a systematic review” by Lauritano et al.
This manuscript reviewed oral health status and the need for oral care in people with dementia, compared to people without dementia. This study also reviewed the relationship between tooth loss due to periodontal disease and cognitive impairment.
The reviewer’s concerns are as follows.
The title includes the word “Oral health-related quality of life”. However, nothing was mentioned or reviewed with respect to oral health related QOL. It is very discouraging that the reference numbers in the text are written in Roman and Arabic style. For me, it is hard to see the Roman style. Please unify. Tables should be reconstructed. For example, in Table 1, the word “Representativeness” is not correctly punctuated. Also there is no explanation for “ + +” ,”+,+” “+,-“, “+” , “-,+” and “+” of Representativeness of exposed cohort, Selection of non exposed cohort, Ascertainment of exposure and etc. Table 2 & 3 list the studies about coronal/root caries, number of remained teeth/retained roots, DMFT, periodontal disease in elderly with and without dementia. These information (number of each group, oral health measures) are repeated in the Table 4-8. Is there any consideration of the systemic diseases with diabetes or functional disabilities affecting oral health status? Line 30: Put “ ) ” after NOS. line 123 Delete “of” . Line 107-111. P,I and C are in bold style, however O is in normal style. In Table 2 & 3, authors name and publication year is normal style, however bold in the other tables. Some of the papers selected for reviewing lack the control group information. Is it worthy for reviewing?